# Diagnostic Value of Conventional PET Parameters and Radiomic Features Extracted from 18F-FDG-PET/CT for Histologic Subtype Classification and Characterization of Lung Neuroendocrine Neoplasms

**DOI:** 10.3390/biomedicines9030281

**Published:** 2021-03-10

**Authors:** Philippe Thuillier, Virginia Liberini, Osvaldo Rampado, Elena Gallio, Bruno De Santi, Francesco Ceci, Jasna Metovic, Mauro Papotti, Marco Volante, Filippo Molinari, Désirée Deandreis

**Affiliations:** 1Nuclear Medicine Unit, Department of Medical Sciences, University of Turin, 10126 Turin, Italy; virginia.liberini@unito.it (V.L.); francesco.ceci@unito.it (F.C.); desiree.deandreis@unito.it (D.D.); 2Department of Endocrinology, University Hospital of Brest, 29200 Brest, France; 3Medical Physics Unit, AOU Città della Salute e della Scienza, 10126 Turin, Italy; orampado@cittadellasalute.to.it (O.R.); egallio@cittadellasalute.to.it (E.G.); 4Biolab, Department of Electronics and Telecomunications, Politecnico di Torino, 10129 Turin, Italy; bruno.desanti@polito.it (B.D.S.); filippo.molinari@polito.it (F.M.); 5Pathology Unit, Department of Oncology, University of Turin, 10126 Turin, Italy; jasna.metovic@unito.it (J.M.); mauro.papotti@unito.it (M.P.); 6Department of Oncology, San Luigi Hospital, University of Turin, 10043 Orbassano, Italy; marco.volante@unito.it

**Keywords:** texture analysis, Radiomics, lung neuroendocrine neoplasm, carcinoid tumor, 18FDG-PET/CT

## Abstract

Aim: To evaluate if conventional Positron emission tomography (PET) parameters and radiomic features (RFs) extracted by 18F-FDG-PET/CT can differentiate among different histological subtypes of lung neuroendocrine neoplasms (Lu-NENs). Methods: Forty-four naïve-treatment patients on whom 18F-FDG-PET/CT was performed for histologically confirmed Lu-NEN (n = 46) were retrospectively included. Manual segmentation was performed by two operators allowing for extraction of four conventional PET parameters (SUVmax, SUVmean, metabolic tumor volume (MTV), and total lesion glycolysis (TLG)) and 41 RFs. Lu-NENs were classified into two groups: lung neuroendocrine tumors (Lu-NETs) vs. lung neuroendocrine carcinomas (Lu-NECs). Lu-NETs were classified according to histological subtypes (typical (TC)/atypical carcinoid (AC)), Ki67-level, and TNM staging. The least absolute shrink age and selection operator (LASSO) method was used to select the most predictive RFs for classification and Pearson correlation analysis was performed between conventional PET parameters and selected RFs. Results: PET parameters, in particular, SUVmax (area under the curve (AUC) = 0.91; cut-off = 5.16) were higher in Lu-NECs vs. Lu-NETs (*p* < 0.001). Among RFs, HISTO_Entropy_log10 was the most predictive (AUC = 0.90), but correlated with SUVmax/SUVmean (r = 0.95/r = 0.94, respectively). No statistical differences were found between conventional PET parameters and RFs (*p* > 0.05) and TC vs. AC classification. Conventional PET parameters were correlated with N+ status in Lu-NETs. Conclusion: In our study, conventional PET parameters were able to distinguish Lu-NECs from Lu-NETs, but not TC from AC. RFs did not provide additional information.

## 1. Introduction

Lung neuroendocrine neoplasms (Lu-NENs) represent a group of rare neoplasms and are classified into four histological subtypes: lung neuroendocrine tumors (Lu-NETs) including typical (TC) and atypical carcinoid (AC) and lung neuroendocrine carcinomas (Lu-NECs) including large-cell neuroendocrine carcinomas (LCNECs) and small-cell neuroendocrine carcinomas (SCLCs) [1]. The prognosis of Lu-NENs is highly related to the histological subtypes. Lu-NECs are associated with a poor prognosis with a 5-year survival rate of 15% in LCNEC and 2% in SCLC, respectively [2], while patients affected by Lu-NETs hold a better prognosis. However, ACs are associated with poorer survival rates compared with TCs (44% and 87%, respectively) [2], with higher rates of lymph node involvement at diagnosis (36% vs. 9%) and distant metastases (26% vs. 4%) [3]. In cases of locally confined disease, Lu-NETs are generally treated with radical surgery and eventual nodal dissection is considered in cases of findings suspected for malignancy at pre-operative staging [4,5]. Thus, correct identification of the different Lu-NENs histological patterns is crucial in the decision-making process [6].

Positron emission tomography (PET)/computed tomography (CT) is recommended to investigate Lu-NENs [7]. 18F-Fluoro-deoxyglucose (FDG) is widely used in Lu-NECs, but also in Lu-NETs [8,9,10], while 68Ga-DOTA-peptides PET/CT is suggested mainly in Lu-NETs (80% of tumor subtypes express somatostatin receptors) [11,12]. Several studies assessed the diagnostic performance of dual 18F-FDG and 68Ga-DOTA-peptides-PET/CT in detecting lung carcinoid [12,13,14,15], suggesting their complementary role in discriminating TC from AC. Nevertheless, both exams are rarely performed before surgery. Grøndahl et al. reported that 18F-FDG-PET/CT was performed in 207 (82%) of 252 patients with Lu-NETs, while 68Ga-DOTATOC-PET/CT was performed preoperatively only in 12% of patients [5]. Intratumoral tumor heterogeneity is one of the hallmarks of malignancy, aggressiveness, treatment response, and prognosis [16]. Indeed, PET radiopharmaceuticals present a different pattern of uptake within the tumor due to differences in the spatial distribution of intratumoral components [17,18,19], such as vascularity, hypoxia, necrosis, proliferation, cellular composition, and inflammation [20,21,22]. Established conventional and volumetric PET parameters, such as standardized uptake value (SUV), metabolic tumor volume (MTV), and total lesion glycolysis (TLG), do not allow for assessment of the heterogeneity of radiotracer uptake in a tumor. Recently, tools have been developed to enable the quantitative description of tumor heterogeneity through image-derived radiomic features (RFs) [23]. Recently, the extraction of RFs from 18F-FDG-PET/CT was applied to characterize the histological pattern and prognosis of non-small cell lung cancers (NSCLCs) [24,25]. To the best of our knowledge, the radiomic approach has not yet been applied to characterize Lu-NENs.

In this study, we have hypothesized that both conventional PET parameters (i.e., SUV-based and volumetric parameters), and RFs extracted from 18F-FDG-PET/CT might allow for more accurate definition of the histologic patterns and phenotypes of Lu-NENs. Therefore, the primary objective of this study was to investigate the diagnostic value of both conventional parameters and RFs to distinguish NETs from NECs and then, TC from AC. The secondary objective was to determine if these parameters are associated with pathological characteristics of tumor aggressiveness (such as mitotic index, presence of necrosis, and Ki-67 index), and TNM stage in Lu-NETs.

## 2. Materials and Methods

### 2.1. Study Design and Inclusion Criteria

This was a retrospective, single-center analysis performed in patients who underwent 18F-FDG-PET/CT in our institution from 1 January 2012 to 1 January 2020. 

Inclusion criteria were: (a) Histologically proven Lu-NEN diagnosis (obtained by core biopsy in 10 samples or surgery in 36 samples), and classified according to the current WHO classification [26]; (b) naïve-treatment patients who underwent 18F-FDG-PET/CT; (c) patients who consented to participate in the study (IRB number: 0099593, code Lung-NET Radiomics_2020)

Exclusion criteria were: (a) 18F-FDG-PET/CT performed after surgery; (b) Cytological data available only.

### 2.2. Clinical Evaluation and Lu-NEN Classification 

All clinical and histological data (age, gender, TNM) [27] were collected. Lu-NENs were classified according to the WHO 2015 classification [26] into TC and AC (grouped as Lu-NETs) and LCNEC and SCLC (grouped as Lu-NECs). Finally, we stratified each lesion from the Lu-NETs group according to histological data: (a) mitosis number (three groups: <2; 2–10; >10/mm^2^); (b) Ki-67 level (three groups: ≤5; >5 and ≤20; >20%) [28]; and (c) TNM staging at the time of diagnosis. 

### 2.3. PET/CT Protocol

All patients underwent FDG PET/CT on the same PET scanner (Philips Gemini Dual-slice EXP scanner, PET AllegroTM system with Brilliance CT scanner, Philips Medical Systems, Cleveland, OH). The median time interval between PET imaging and biopsy was 1.5 months (0–6 months). In accordance with the procedure guidelines for PET imaging [29], the injected 18F-FDG activity was 242 +/− 52 MBq (range, 148–393 MBq). After 60 min of uptake and following free-breathing CT acquisition for attenuation correction from the vertex of the skull to the mid-thigh (5 mm slice, 40 mAs and 120 kVp), PET data were acquired in 3-dimensional (3D) mode, covering the same anatomical region of the CT, with 2.5 min per bed position and 6–8 bed positions per patient. The PET scans were reconstructed by the ordered subset expectation maximization (OSEM) algorithm (3D-RAMLA), with the following settings: 4 iterations, 8 subsets, and a field of view (FOV) of 576 mm. For all reconstructions, the matrix size was 144 × 144 voxels, resulting in isotropic voxels of 4.0 × 4.0 × 4.0 mm^3^. All acquisitions were corrected for attenuation (using the corresponding CT image), as well as for scatter and random coincidences.

### 2.4. Conventional PET Parameters and RFs Extraction

All primary lesions were analyzed using LIFEx v. 6.0 (IMIV/CEA, Orsay, France) and segmented manually by two operators (VL and PT). Manual segmentation for FDG-avid lesions was performed on PET images. Manual segmentation for non-avid lesions was performed on the coupled CT images. In the volume of interest (VOI), intensities of 18F-FDG uptake were resampled using absolute intensity rescaling factors of 0–20 of the SUV of the VOI (64 bins, 0.32 fixed bin width). The number of grey levels was set to 64 based on the results of previous studies regarding the RFs’ robustness in 18F-FDG-PET/CT [30,31,32]. We used native voxel sizes (4 × 4 × 4 mm). A total of forty-five conventional PET parameters and RFs (i.e., histogram-based, shape-based, and textural features) were extracted, using LIFEx in agreement with the Imaging Biomarker Standardization Initiative (IBSI) description [33]:

Four conventional and volumetric PET parameters (SUVmax, SUVmean, metabolic tumor volume (MTV) and total lesion glycolysis (TLG)). The MTV represents the volume of the segmented VOI; TLG is calculated by multiplying the MTV of each lesion with its corresponding SUVmean value [29];Forty-one RFs: six descriptors of the image intensity histogram: HISTO_Skewness (asymmetry), HISTO_Kurtosis (flatness), HISTO_ExcessKurtosis (peakedness), HISTO_Energy (uniformity), HISTO_Entropy_log2, and _log10 (randomness); three shape-based features that describe the shape of the VOI: SHAPE_Sphericity, SHAPE_Surface (mm^2^), and SHAPE_Compacity; thirty-two textural features: (a) seven features from the grey-level co-occurrence matrix (GLCM): describing the correlation between a pair of voxels in 13 directions of a three-dimensional space; (b) eleven features from the grey-level run-length matrix (GLRLM): describing the number and length of the run with a certain level of grey in 13 directions of a three-dimensional space; (c) eleven features from the grey-level zone length matrix (GLZLM): describing the number and size of the zone with a certain level of grey in 13 directions of a three-dimensional space; and (d) three features from the neighborhood grey-level different matrix (NGLDM): describing the difference between a voxel and its connected neighbors.

### 2.5. Statistical Analysis

Quantitative variables are reported as mean ± standard deviation (SD), or as median and interquartile range, depending on their distribution. The categorical variables are expressed as a percentage. The continuous quantitative variables were compared by non-parametric tests (Mann–Whitney U or Kruskal–Wallis).

#### 2.5.1. Conventional PET Parameters

For conventional PET parameters, we performed Receiver Operating Characteristics (ROC) analysis to assess the diagnostic performance of each of the conventional parameters to classify each lesion. The Youden index was used to find each variable’s best cut-off point. The area under the curves (AUC), sensitivity, specificity, and accuracy were measured.

#### 2.5.2. Radiomics Features Selection

The reproducibility of RFs between the two operators was assessed by Intra-class Correlation Coefficients (ICC) using a two-way mixed effects model. ICC values lie between 0 and 1 and a ICC > 0.9 was retained to consider the RFs sufficiently robust [34,35].

Among RFs with ICC > 0.9, the Mann–Whitney test was used to compare the RFs between each group of patients and select RFs able to discriminate the 2 groups of patients (Lu-NECs versus Lu-NETs and then TC versus AC). 

To reduce the potential redundancy among the RFs extracted in this study, the most useful predictive parameters were selected using the least absolute shrinkage and selection operator (LASSO) logistic regression model [36,37]. Then, a Pearson correlation analysis was performed between conventional PET parameters and RFs extracted from the LASSO regression. Then, a multivariate logistic regression analysis was performed to calculate a radiomic signature with the selected RFs. Finally, a ROC curve was used to illustrate the diagnostic performance of the model to predict the histological pattern.

All *p*-values were obtained by the 2-sided exact method at the conventional 5% significance level. Statistical analysis was performed using XLSTAT 2019 and R software (version 1.1.419) [38].

## 3. Results

### 3.1. Patient Population

Forty-four patients and 46 lesions histologically proven for Lu-NENs were enrolled. The main characteristics and clinical details of the patients are represented in Table 1. Among the 44 patients included in the study, only three patients also underwent 68Ga-DOTATOC-PET/CT before core biopsy/surgery.

### 3.2. Tumor Subtypes Classification According to Histology

#### 3.2.1. Lu-NECs vs. Lu-NETs

All conventional PET and volumetric parameters resulted higher in Lu-NECs compared with Lu-NETs (*p* < 0.001; Table 2). At ROC curve analysis, the best cut-off values to distinguish between Lu-NECs and Lu-NETs for SUVmax, SUVmean, MTV, and TLG were: 5.16 (Se = 0.84, Sp = 0.85, and Acc = 0.85; AUC = 0.91), 3.69 (Se = 0.79, Sp = 0.96, and Acc = 0.89; AUC = 0.91), 8.96 (Se = 0.68, Sp = 0.85, and Acc = 0.78; AUC = 0.8), and 38.67 (Se = 0.79, Sp = 0.89, and Acc = 0.85; AUC = 0.86), respectively (Figure 1 and Table 3). One AC presented a SUVmean = 4.5 (SUVmax = 8.8), while one SCLC and three LCNECs presented low SUVmean values (SUVmean = 3.46 and SUVmean of 1.43, 2.24, and 2.70, respectively).

Regarding RFs, the inter-observer agreement between the two operators showed that 30/42 RFs (73.2%) were highly robust (6/6 histogram, 2/3 shape, and 22/32 TFs; ICC > 0.9). Among the 30 RFs with ICC > 0.9, 26 were significantly associated with histological subtypes (all *p*-values < 0.05). Then, after LASSO implementation, HISTO_Entropy_log10 was selected as the most predictive RF, with no significant advantages in adding more RFs to the model (Figure 2). At the ROC curve analysis, HISTO_Entropy_log10 showed an AUC of 0.90, with a cut-off value of 0.94 (Se = 0.90, Sp = 0.78, and Acc = 0.83; Figure 1 and Table 3). Finally, HISTO_Entropy_log10 was significantly positive correlated with SUVmax and SUVmean (r = 0.95 and 0.94, respectively). After combining both conventional parameters and HISTO_Entropy_log10 in a logistic regression model, no significative advantages were found with respect to the model with only SUVmean to predict the histological subtypes.

Plot of area under the curve (AUC) values versus the logarithm of the penalization parameter lambda (λ). Log (λ) = −3.63, with λ = 0.027 was chosen.

#### 3.2.2. Lu-NETs: Typical vs. Atypical Carcinoid Tumors

None of the conventional parameters were able to distinguish between TCs and ACs. Median SUVmax (3.62 vs. 2.8; *p* = 0.68) and SUVmean (2.37 vs. 1.62; *p* = 0.65) were not statistically different between TC and AC (Table 2). None of the RFs were significantly different among TC and AC.

### 3.3. Comparison between Histological Data and TNM Status in Lu-NETs

Considering the lack of significance for RFs in the Lu-NET group, only conventional PET parameters were considered in evaluating potential associations with pathological parameters and TNM staging.

Regarding histological data, none of the PET parameters evaluated were associated with mitotic count or the presence of the necrosis (*p* > 0.05). Stratifying TC and AC according to Ki-67 level, two TCs presented Ki-67 > 5%, while one AC had a Ki-67 ≤ 5% and two ACs presented a Ki-67 > 20. SUVmax and SUVmean showed a positive trend with Ki-67, without showing statistical significance (*p* = 0.05 and 0.07, respectively, Figure 3). The two patients with AC and a Ki-67 index > 20% had much higher SUV values than TC and AC with Ki-67 <20% (Figure 3). Using Ki-67 as a continuous variable, there was an association between Ki-67 and SUVmax (R = 0.52, *p* = 0.007).

Regarding TNM status, SUVmax, MTV, and TLG of the primary lesion were significantly associated with N+ status (*p* < 0.05; Figure 4). At the ROC curve analysis, SUVmax proved to be the most accurate predictor of N+ status (AUC = 0.78; *p* = 0.004) with sensitivity = 0.67, specificity = 0.82, and accuracy = 0.77, applying a cut-off of 4.11. In the Lu-NET subgroup, only three patients were M+ at the time of diagnosis, and conventional and volumetric parameters showed a trend of correlation with M+ (*p* = 0.08 for MTV and TLG; Figure 5).

## 4. Discussion

In this study, we assessed the association between both conventional PET parameters and RFs extracted from 18F-FDG-PET/CT with histopathological subtypes of Lu-NENs. Our study showed the good diagnostic performance of conventional parameters in the identification of Lu-NETs vs. Lu-NECs, allowing us to discriminate between these two groups of patients, while our results did not suggest any potential additional value of the RFs. Regarding our subgroup analysis considering Lu-NETs, we did not observe any difference between conventional PET parameters or RFs and TC and AC classification. On the other hand, higher values of conventional and volumetric PET parameters of the primary lesion were able to predict N+ status.

In the discrimination between Lu-NETs and Lu-NECs, we observed that SUVmax and SUVmean were more discriminating parameters, even though some Lu-NECs could have low SUVmax/ SUVmean values. These results appear to be widely explained by the low tumor volume in our cohort. In a recent study including 31 LCNEC patients, the mean SUVmax value in this population was 9.0 ± 3.8 (range 2.3–17.2; median 8.9) vs. 12.4 ± 6.3 (range 2.3–22.6; median 12.7) in our population. These findings are consistent with our results, showing the potential low FDG-avidity in Lu-NECs [39].

Regarding the diagnostic performance of RFs, we found that RFs do not provide additional information allowing us to discriminate Lu-NECs and Lu-NETs. In our methodology, we used an absolute intensity resampling of 0–20 of the SUV with a fixed number of 64 bins. These grey levels and the absolute intensity resampling were based on the results of previous studies assessing RFs’ robustness in 18F-FDG-PET/CT [32,40]. Otherwise, SUV 0–20 corresponds to the typical range of tumor SUVs encountered in Lu-NENs [6,39]. To avoid the redundancy of RFs, we used the “glmnet” package in R to perform a LASSO logistic regression to select the most predictive RFs for the classification, as recently reported [36,37]. Using this methodology, only HISTO_Entropy_log10 was selected by the LASSO regression, but it was highly correlated to conventional PET parameters. These results may be explained by the characteristics of our Lu-NENs population. First, even if several NETs might present high FDG-avidity, while several NECs hold low FDG-avidity, the broad difference in SUV values in NECs compared with NETs could lead to a broader difference in RF values between the two groups [32]. Moreover, the volume of Lu-NENs in our cohort was low (median value of MTV = 7.52 mL) and several studies reported that a radiomic approach does not provide additional information when the lesion metabolic volume is lower than 10 mL [41]. In addition, we performed a complementary analysis excluding the three patients with SCLC and found similar results. After LASSO implementation, HISTO_Entropy_log10 was also selected as the most predictive RF. At the ROC curve analysis, HISTO_Entropy_log10 showed a similar AUC of 0.91, but did not provide additional value comparing to the conventional PET parameters (Appendix A).

In the Lu-NET groups, we did not find any difference between TC and AC among all parameters when considering SUV parameters or other RFs. When related to the radiomic approach, these results could also be explained by the same reasons as for the Lu-NETs and Lu-NECs (i.e., no difference and a low SUV value in this case and a small size of lesions). Regarding the diagnostic performance of conventional parameters, these results appear to be partially inconsistent with the literature. Indeed, a recent study highlighted that AC had a higher median SUVmax value in comparison with TC (median SUVmax of 6.3 (range 1.6–29) vs. 3.7 (range 1.2–13.3), respectively, *p* = 0.03) [42]. A recent meta-analysis found similar results with higher SUVmax values for AC compared with TC (SUVmax = 6.0 (range 1.7–14.5) vs. SUVmax = 3.7 (range 0.8–16.0), *p* < 0.05). Nevertheless, several studies found similar results to ours with no differences between SUV parameters in TC and AC [43,44]. The size of the lesion might have affected these findings and may partially explain these discrepancies [45]. Indeed, SUV is widely affected by the partial volume effect and is positively correlated with SUVmax in Lu-NETs [44,45] and AC present in a higher volume than TC in the literature, which was not the case in our study. Thus, 18F-FDG-PET/CT alone appears inadequate to distinguish TC from AC. Dual tracer PET/CT combining both 18F-FDG and 68Ga-DOTA peptides appears to be the best approach to detect and characterize Lu-NETs. Using this approach, a recent meta-analysis suggested the usefulness of the SUVmax DOTA to FDG ratio to discriminate TC and AC with excellent diagnostic performance (AUC = 0.94 with Se = 0.89 and Sp = 1.00) [6].

Regarding histopathological data and stratifying TC and AC according to the Ki-67 level, we found that two TCs presented Ki-67 > 5%, while one AC had a Ki-67 ≤ 5% and two other ACs presented a Ki-67 > 20% in our cohort. These results are consistent with the literature and showed that the Ki-67 distribution in Lu-NETs can be variable, and the optimal cut-off value to discriminate TC from AC is still debated [46,47]. We highlighted that SUV and Ki-67% showed a positive trend of correlation. The two patients with AC and a Ki-67 index > 20% had much higher SUV values than TC and AC with Ki-67 < 20%, although the low number of cases limits the statistical significance in our cohort. Thus, the SUVmax may be a relevant criterion for identifying lung tumors with high Ki-67, especially those with Ki-67 > 20 with a poor prognosis [48].

Regarding TNM status, SUVmax, MTV, and TLG were significantly associated with N+ status. Several studies reported a poor sensitivity and specificity of preoperative 18F-FDG-PET/CT to diagnose lymph node involvement [44]. In our study, we analyzed primary lesions only, observing that PET parameters are good predictors of N+ status.

This study is not exempt from limitations. First, this analysis was performed in a single center and these results cannot be extrapolated to other centers. Then, we applied a manual segmentation and the impact of the use of other segmentation methods was not evaluated in this study. However, the majority of the RFs were reproducible between the two operators. Finally, the radiomic approach appears to be very limited in this population, considering the population characteristics (SUV value and volume differences according to the histological subtypes).

## 5. Conclusions

In this study, we observed the good diagnostic performance of conventional PET parameters of 18F-FDG-PET/CT used to discriminate between Lu-NECs and Lu-NETs. However, our results did not show a potential additional value for RFs. Furthermore, in our study no difference was observed between TC vs. AC subtypes, either considering conventional PET parameters or RFs. Nevertheless, a trend in the association between SUV values and the Ki-67 index, with special reference to the subgroup of cases with Ki-67 values exceeding 20%, has been found along with an association between SUV and N+ status. For this reason, conventional PET parameters might be applied to evaluate the tumor aggressiveness and to predict lymph node involvement in Lu-NETs. These preliminary results need to be validated in larger cohorts, even applying different methodologies, to assess the possible contribution of the radiomic analysis in PET imaging within this clinical setting.

## Figures and Tables

**Figure 1 biomedicines-09-00281-f001:**
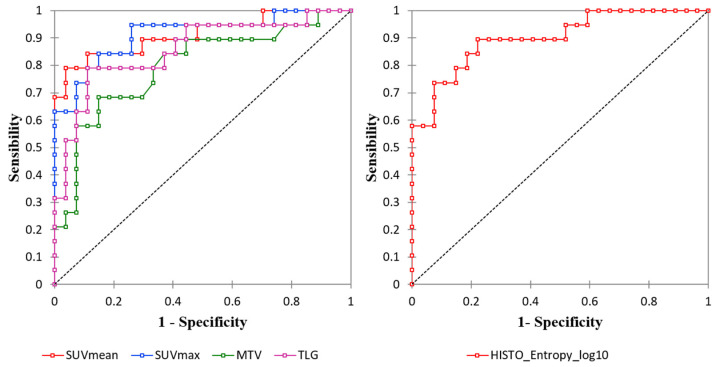
Receiver operating characteristics (ROC) curve analysis of the conventional and volumetric PET parameters (**left**) and Histo_Entropy_log10 (**right**).

**Figure 2 biomedicines-09-00281-f002:**
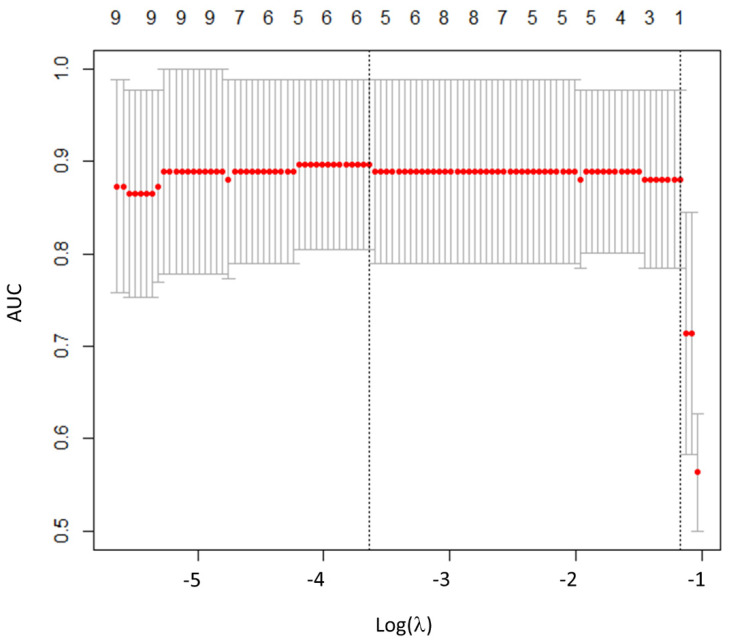
Radiomics features selection using least absolute shrinkage and selection operator (LASSO).

**Figure 3 biomedicines-09-00281-f003:**
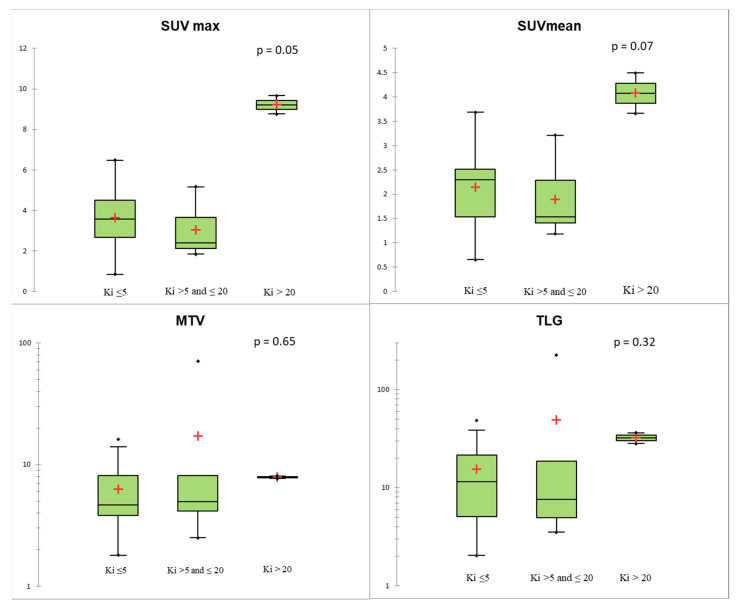
Conventional and volumetric parameters according to Ki-67 level using the Kruskal–Wallis test. The red crosses correspond to the mean value. The central horizontal bars correspond to the median value. The lower and upper limits of the boxes are the first and third quartiles. The error bars are the first decile and the ninth decile; the rhombuses are the minimum and maximum for each parameter.

**Figure 4 biomedicines-09-00281-f004:**
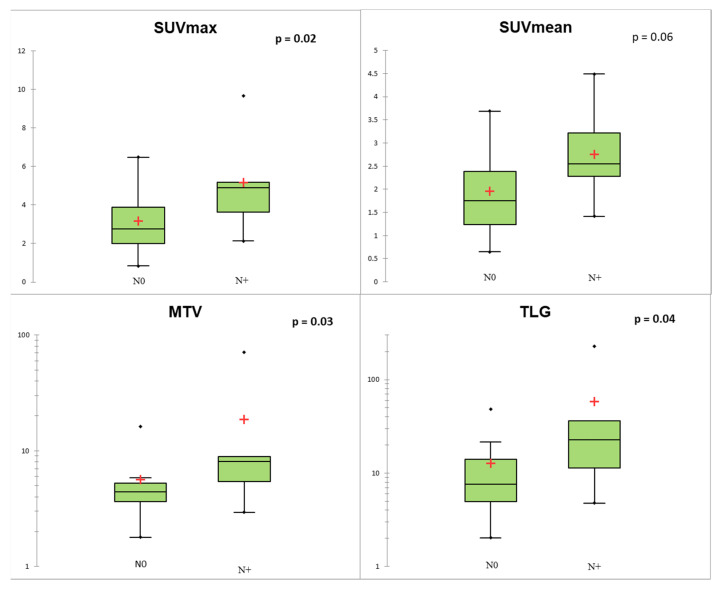
Conventional and volumetric parameters according to N status. The red crosses correspond to the mean value. The central horizontal bars correspond to the median value. The lower and upper limits of the boxes are the first and third quartiles. The error bars are the first decile and the ninth decile; the rhombuses are the minimum and maximum for each parameter.

**Figure 5 biomedicines-09-00281-f005:**
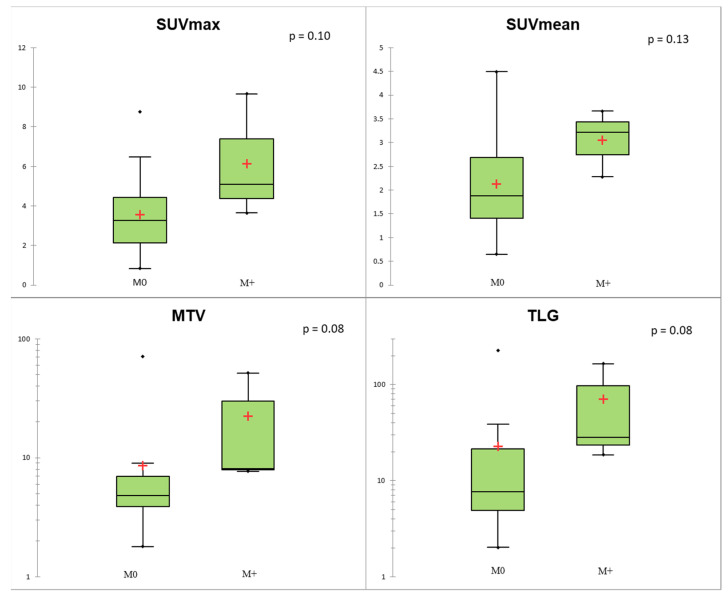
Conventional and volumetric parameters according to M status. The red crosses correspond to the mean value. The central horizontal bars correspond to the median value. The lower and upper limits of the boxes are the first and third quartiles. The error bars are the first decile and the ninth decile; the rhombuses are the minimum and maximum for each parameter.

**Table 1 biomedicines-09-00281-t001:** Clinical and histological characteristics of the cohort.

Characteristics	Value
Sex, n (%)	
Male	23 (52.3)
Female	21 (47.7)
Age (years); moy (DS)	62.8 (10.1)
Lesion side, n (%)	
Right	29 (63)
Left	17 (37)
Size (mm); moy (DS)	30.5 (16)
T-classification, n (%)	
1	17 (37.0)
2	11 (23.9)
3	8 (17.4)
4	0 (0)
X	10 (21.7)
N-classification, n (%)	
0	26 (56.5)
1	20 (43.5)
M-classification, n (%)	
0	41 (89.1)
1	5 (10.9)
Histological subtypes	
TC	15 (32.6)
AC	11 (23.9)
TC or AC	1 (2.2)
LCNEC	16 (34.8)
SCLC	3 (6.5)
Mitosis/mm^2^ (n = 35)	
<2	15 (42.8)
2–10	10 (28.6)
10/mm^2^	10 (28.6)
Ki67% (n = 45)	
≤5%	15 (33.3)
>5 and ≤20%	10 (22.2)
>20%	20 (44.4)

**Table 2 biomedicines-09-00281-t002:** Comparison between conventional and volumetric PET parameters and HISTO_Entropy_log10 in neuroendocrine carcinomas (NECs) versus neuroendocrine tumors (NETs) and in typical carcinoid (TC) versus atypical carcinoid (AC).

		Lung-NETs *			Lung-NECs		*p* Value	*p* Value
	Total (n = 27) *	TC (n = 15)	AC (n = 11)	Total (n = 19)	LCNEC (n = 16)	SCLC (n = 3)	(NETs vs. NECs)	(TCs vs. ACs)
SUVmax	3.52 (0.84–9.66)	3.62 (0.84–6.48)	2.81 (1.84–9.66)	10.71 (2.27–22.65)	12.74 (2.27–22.65)	10.52 (4.60–15.66)	<0.001	0.68
SUVmean	2.23 (0.65–4.49)	2.37 (0.65–3.69)	1.62 (1.13–4.49)	6.14 (1.43–10.09)	6.49 (1.43–10.09)	5.97 (3.47–7.72)	<0.001	0.65
MTV	4.93 (1.79–70.91)	4.93 (2.62–70.91)	4.93 (1.79–51.39)	27.01 (2.88–376.38)	27.97 (2.88–376.38)	9.09 (4.16–27.78)	<0.001	0.51
TLG	11.38 (2.03–226.1)	11.68 (2.83–226.11)	7.69 (2.03–70.91)	167.74 (4.11–2578.64)	185.00 (4.11–2578.64)	54.21 (14.43–214.61)	<0.001	0.45
HISTO_Entropy_log10	0.77 (0.27–1.26)	0.77 (0.27–1.12)	0.77 (0.49–1.26)	1.3 (0.68–1.71)	1.40 (0.68–1.71)	1.30 (0.76–1.48)	<0.001	0.96

* 1 lesion classified in Lung-NETs according to biopsy but was not conclusive to make the diagnosis of TC or AC.

**Table 3 biomedicines-09-00281-t003:** Diagnostic performance of conventional and volumetric PET parameters and HISTO_Entropy_log10 in NECs versus NETs.

Parameters	AUC	CI 95%	*p* Value	Cut-Off Value	Se (%)	Sp (%)	Acc (%)
**SUVmax**	0.91	(0.82–1.00)	<0.001	5.16	84.2	85.2	84.8
**SUVmean**	0.91	(0.82–1.00)	<0.001	3.69	78.9	96.3	89.1
**MTV (ml)**	0.80	(0.66–0.93)	<0.001	8.96	68.4	85.2	78.3
**TLG (g)**	0.86	(0.74–0.97)	<0.001	38.67	78.9	88.9	84.8
**HISTO_Entropy_log10**	0.90	(0.81–0.99)	<0.001	0.94	89.5	77.8	82.6

AUC: Area under the curve; 95%CI: 95% Confidence interval; Se: sensibility; Sp: specificity; Acc: Accuracy.

## Data Availability

The data presented in this study are available on request from the corresponding author.

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
