# Peer review of "Diagnostic Value of Conventional PET Parameters and Radiomic Features Extracted from 18F-FDG-PET/CT for Histologic Subtype Classification and Characterization of Lung Neuroendocrine Neoplasms"

_biomedicines, 2021, doi:10.3390/biomedicines9030281_

Round 1
Reviewer 1 Report
This is an interesting and well-written paper.
This are my comments/questions:
- in the first paragraph, there seems to be an error in the sentence: However, ACs are associated with poorer survival rates compared to TCs (87% and 44%, respectively). It should probably be (44% and 87%, respectively).
- My main concern with the results is the Lu-NECs group. As mentioned in the discussion LCNEC have potentially low FDG-avidity. Whereas, SCLC usually has high FDG-avidity. There are only 3 patients in this cohort with SCLC. Do the results still apply when only LCNEC are taken into account? This would be a more interesting result in the clinical setting. Even though the size of the group of SCLC is small, it would be relevant to know the SUVmax, SUVmean, MTV and TLG for both groups.
- It is explained that the VOI of non-avid lesions is drawn according to the CT. It is not explained how the VOI of FDG-avid lesions are determined: according to the CT or with a % SUV threshold?
Author Response
Dear reviewer,
We thank you for reviewing our article entitled “Diagnostic value of conventional PET parameters and radiomic features extracted from 18F-FDG-PET/CT for histologic subtype classification and characterization of lung neuroendocrine neoplasms”
According to your comments, we have carefully revised our manuscript. Please find our point by point response.
Comments and Suggestions for Authors
This is an interesting and well-written paper.
Dear reviewer, we thanks you for this positive comments.
This are my comments/questions:
- in the first paragraph, there seems to be an error in the sentence: However, ACs are associated with poorer survival rates compared to TCs (87% and 44%, respectively). It should probably be (44% and 87%, respectively).
Dear Reviewer, we corrected this mistake.
- My main concern with the results is the Lu-NECs group. As mentioned in the discussion LCNEC have potentially low FDG-avidity. Whereas, SCLC usually has high FDG-avidity. There are only 3 patients in this cohort with SCLC. Do the results still apply when only LCNEC are taken into account? This would be a more interesting result in the clinical setting. Even though the size of the group of SCLC is small, it would be relevant to know the SUVmax, SUVmean, MTV and TLG for both groups.
Dear Reviewer, we performed a complementary analysis excluding the 3 patients with SCLC and found similar results. After LASSO implementation, HISTO_ Entropy_log10 was also selected as the most predictive RF. At the ROC curve analysis, HISTO_ Entropy_log10 showed a similar AUC of 0.91.
We added these precisions in the discussion “In addition, we performed a complementary analysis excluding the 3 patients with SCLC and found similar results. After LASSO implementation, HISTO_ Entropy_log10 was also selected as the most predictive RF. At the ROC curve analysis, HISTO_ En-tropy_log10 showed a similar AUC of 0.91 but did not provide additional value com-paring to the conventional PET parameters (supplementary figure 1 and 2)”.
In addition we had the values of PET parameters and HISTO_ Entropy_log10 for LCNEC and SCLC in the table 2.
- It is explained that the VOI of non-avid lesions is drawn according to the CT. It is not explained how the VOI of FDG-avid lesions are determined: according to the CT or with a % SUV threshold?
Dear Reviewer, for FDG-avid lesions we performed a manual segmentation using PET imaging. We added in the manuscript “Manual segmentation for FDG-avid lesion was performed on PET images”

Reviewer 2 Report
- The current manuscript investigated the conventional PET parameters of 18F-FDG-PET/CT could discriminate between Lu-NECs from Lu-NETs. The best cut-off values for SUVmax, SUVmean, MTV and TLG were obtained. However, can these key parameters be presented in a table?
- Many abbreviations should also be defined at first mention, such as PET, Lu-NETs, Lu-NECs and SCNEC.
- p2 line104: is it a typo between 242 and 52? Is it “±”?
- Radiomic feature exhibited no significant difference between TC and AC. What was the time interval between PET imaging and biopsy? Please add the information in Materials and Methods. Could there be any discrepancy induced by tumor progression?
- Please check Figure 2, 3 and 4 and descript what are the error bars in the captions.
Author Response
Dear reviewer,
We thank you for reviewing our article entitled “Diagnostic value of conventional PET parameters and radiomic features extracted from 18F-FDG-PET/CT for histologic subtype classification and characterization of lung neuroendocrine neoplasms”
According to your comments, we have carefully revised our manuscript. Please find our point by point response.
Comments and Suggestions for Authors
- The current manuscript investigated the conventional PET parameters of 18F-FDG-PET/CT could discriminate between Lu-NECs from Lu-NETs. The best cut-off values for SUVmax, SUVmean, MTV and TLG were obtained. However, can these key parameters be presented in a table?
Dear reviewer, we added this table in the revised manuscript as table 3.
- Many abbreviations should also be defined at first mention, such as PET, Lu-NETs, Lu-NECs and SCNEC.
Dear reviewer, we checked the abbreviations in the abstract and the main text.
- p2 line104: is it a typo between 242 and 52? Is it “±”?
Dear Reviewer, we corrected this mistake.
- Radiomic feature exhibited no significant difference between TC and AC. What was the time interval between PET imaging and biopsy? Please add the information in Materials and Methods. Could there be any discrepancy induced by tumor progression?
Dear reviewer, the median time interval between PET imaging and biopsy was 1.5months in our cohort (0-6months). We added this precision in the materials and methods section “The median time interval between PET imaging and biopsy was 1.5 months (0-6months)”. Regarding the question about discrepancy induced by tumor progression, we don't think it can be a problem when the delay between the PET and the biopsy/surgery is <6months in NETs.
- Please check Figure 2, 3 and 4 and descript what are the error bars in the captions.
Dear reviewer, we added the full description of the figure in the legend. “The red crosses correspond to the mean value."The central horizontal bars correspond to the median value. The lower and upper limits of the boxes are the first and third quartiles. The error bar are the first decile and the ninth decile and the rhombuses are the minimum and maximum for each parameter”.

Round 2
Reviewer 1 Report
Thank you for this corrected version of your manuscript.
My questions have been answered and I have no further comments.
Reviewer 2 Report
All my questions have been well addressed. I don't have further comments.